# Excavation of Molecular Subtypes of Endometrial Cancer Based on DNA Methylation

**DOI:** 10.3390/genes13112106

**Published:** 2022-11-13

**Authors:** Yujie Liu, Yue Gu, Mengyan Zhang, Jiaqi Zeng, Yangyang Wang, Hongli Wang, Xueting Liu, Sijia Liu, Zhao Wang, Yuan Wang, Le Wang, Yunyan Zhang

**Affiliations:** 1Department of Gynecological Radiotherapy, Harbin Medical University Cancer Hospital, Harbin 150081, China; 2Computational Biology Research Center, School of Life Science and Technology, Harbin Institute of Technology, Harbin 150001, China

**Keywords:** endometrial cancer, DNA methylation, TCGA, molecular subtypes

## Abstract

Tumor heterogeneity makes the diagnosis and treatment of endometrial cancer difficult. As an important modulator of gene expression, DNA methylation can affect tumor heterogeneity and, therefore, provide effective information for clinical treatment. In this study, we explored specific prognostic clusters based on 482 examples of endometrial cancer methylation data in the TCGA database. By analyzing 4870 CpG clusters, we distinguished three clusters with different prognostics. Differences in DNA methylation levels are associated with differences in age, grade, clinical pathological staging, and prognosis. Subsequently, we screened out 264 specific hypermethylation and hypomethylation sites and constructed a prognostic model for Bayesian network classification, which corresponded to the classification of the test set to the classification results of the train set. Since the tumor microenvironment plays a key role in determining immunotherapy responses, we conducted relevant analyses based on clusters separated from DNA methylation data to determine the immune function of each cluster. We also predicted their sensitivity to chemotherapy drugs. Specific classifications of DNA methylation may help to address the heterogeneity of previously existing molecular clusters of endometrial cancer, as well as to develop more effective, individualized treatments.

## 1. Introduction

Endometrial cancer (EC) is one of the most common malignancies in the female reproductive system [1]. Each year, more than 100,000 women worldwide develop endometrial cancer, and more than 40,000 die from it [2]. The incidence of EC has been on the rise worldwide over the past few decades, and is expected to continue to rise in the coming decades, especially in low- and middle-income countries [3,4]. Despite medical and surgical treatments, outcomes associated with late-stage or high-risk EC are poorer [5,6]. Therefore, the correct understanding of the disease and appropriate cluster classification are essential for the selection of appropriate adjuvant therapy.

The Cancer Genome Atlas (TCGA) identified four genomic subgroups: POLE ultramutated (POLE), microsatellite instability (MSI) hypermutated, copy-number low (CN low), and copy-number high (CN high) [7]. Although genetic alterations, such as mutations and copy number changes, can affect the development of cancer, epigenetic changes in DNA methylation also play an important role in the development of cancer. TCGA classification has greatly advanced our understanding of the molecular diversity and associated prognostic impact of endometrial cancer, but its clinical applicability in refining surgical staging, guiding adjuvant therapy, and post-treatment monitoring remains limited [8]. Epigenetic mechanisms regulating gene activity have attracted much attention in the post-genomic era [9]. Epigenetic changes can occur in the early or late stages of tumor progression [10]. DNA methylation is one of the most deeply studied epigenetic modifications in mammals [11]. Functionally, hypermethylated CpG islands can silence the transcription of various tumor suppressor genes, while hypomethylated CpG islands can activate oncogenes [12]. Thus, specific DNA methylation traits associated with pathogenesis may ultimately serve as useful biomarkers for disease diagnosis, prognosis, disease surveillance, or prediction of treatment response [13].

Endometrial cancer is a heterogeneous disease with an overall 5-year survival rate of about 80% [14]. EC is usually diagnosed by biopsy, in which a small piece of tumor tissue is aspirated through the endometrium, which has good sensitivity and specificity for the diagnosis of cancer. However, in heterogeneous tumors, there may be a small number of cells that are not taken up, and these cells may have a certain impact on diagnosis, prognosis, and treatment [15]. Evidence suggests that aberrant DNA methylation, which is associated with loss of expression of various key genes, can cause extensive alterations in endometrial tumors [16]. In our study, many of these specific CpG sites have previously been reported to be associated with endometrial cancer. For example, the p73 gene is a member of the p53 family, and is involved in signaling pathways such as apoptosis [17]. Lai, H.-C. et al. found that epigenetic hypermethylation of the AJAP1 gene is a valuable biomarker for identifying undetected EC in atypical hyperplasia [18]. Li, L. showed that silencing dSCAM-AS1 could promote apoptosis and inhibit proliferation of EC cells in vitro [19]. Kato, H. et al. identified and validated DCC genes, which regulate normal endometrial cell growth and are used as tumor suppressor genes in EC [20].

In this study, we screened out 4870 potential prognostic methylation sites based on EC DNA methylation data using univariate COX regression analysis and multivariate COX regression analysis. According to the prognostic methylation site, three different prognostic EC clusters were identified by the non-negative matrix decomposition clustering method. Subsequently, we found 246 specific high and hypomethylation sites corresponding to 161 genes that define specific DNA methylation clusters in endometrial cancer. These sites can be seen as targets of precision medicine and biomarkers for diagnosing endometrial cancer. In addition, using these specific CpG sites as features of prognostic models, the test set dataset can be distinguished into different prognostic clusters that are consistent with the train set classification results. Next, we further explore the significance of these three clusters through immune infiltration and chemotherapy drug sensitivity analysis. The results suggest that macrophages may be potential targets for endometrial cancer treatment, that patients with cluster 1 may benefit from immune checkpoint inhibitor therapy, and that cluster 3 may be more sensitive to commonly used chemotherapy drugs. We will provide more therapeutic targets for endometrial cancer by looking for specific molecular markers for each subtype.

## 2. Materials and Methods

### 2.1. Pretreatment and Data Selection of Endometrial Cancer

An array of 482 examples of endometrial cancer samples of Illumina Infinium HumanMethylation450 bead chips was downloaded from the UCSC Xena (https://xenabrowser.net/datapages/, accessed on 21 January 2022) database. β values represent the methylation level of each probe, ranging from 0 to 1 and corresponding to unmethylated and fully methylated levels, respectively. In this sample, more than 70% of the missing data probes were removed, and NAs probes were estimated using k-nearest neighbors (knn). Unstable genomic sites, including single nucleotide polymorphisms and CpGs in sex chromosomes, were removed. The definition of the promoter region is 2 kb upstream to 0.5 kb downstream from the transcription start point. Since the DNA methylation sites in the promoter region have a strong influence on gene expression, we chose CpGs in the promoter region. 

Data from 583 examples of endometrial cancer transcriptome RNA-Seq data were downloaded from the UCSC Xena database for pre-processing, and quantified as RNA-Seq by expectation maximization (RSEM). Zero-valued entries were replaced by the minimal positive value of the dataset. The expression values were normalized with a logarithmic transformation (base 2). Data on four types of TCGA endometrial cancer were downloaded from previously published reports [21].

We deleted the data from the poor sample and, finally, selected 422 examples of endometrial cancer data with both expression profiles and DNA methylation profiles for analysis. We randomly divided the TCGA dataset into train sets and test sets in a 7:3 ratio column.

### 2.2. Difference Analysis and Determination of Classification Characteristics by COX Proportional Risk Regression Model

The *t*-test was used to determine whether the means of the two types of samples are different, and is suitable for data that satisfy normality, variance homogeneity, and low sample content. We analyzed the difference in the methylation dataset of 315 samples by *t* test in order to obtain the significant *p*-value and the corrected FDR value. Identification of significantly different methylation sites where *p* < 0.01 at thresholds and FDR values are less than 0.01 is considered significant.

The purpose of this study was to obtain a molecular cluster of endometrial cancer. Therefore, we classified CpG sites that have a significant impact on survival. First, a one-way COX proportional risk regression model was established according to the methylation level, age, grade, and pathological stage of each CpG site. Significant CpG sites from the univariate COX risk proportional regression model were then placed into the multivariate COX proportional risk regression model. Since age is significant in univariate COX risk regression, it was used as a covariate. Finally, CpG sites that were still significant were used as classification features. COX proportional hazard models were fitted with methylation levels of CpGs using the coxph function in survival package R, and the clinical data (age) was used as the covariate in multivariate analysis. For each CpG *i*, the variable COX proportional risk regression model formula concept is as follows:(1)$$h{\left(t,x\right)}_{i}=h0\left(t\right)exp\left(\beta_{\mathrm{methy}}{\mathrm{methy}}_{i}+\beta_{\mathrm{age}}\mathrm{age}\right)$$

Among them, methy*_i_* is the vector of CpG *i* methylation samples in the sample, age represents the vector of case age, and β_methy_ and β_age_ are regression coefficients. The *p*-value of the COX regression coefficient is adjusted by the incidence of multiple comparisons.

### 2.3. Nonnegative Matrix Decomposition Clustering to Obtain Molecular Subtypes Associated with Endometrial Cancer Prognosis

Nonnegative matrix factorization (NMF) clustering analysis using the “NMF” packet in R was used to determine the DNA methylation cluster of endometrial cancer. Specifically, NMF was applied to methylation matrix A containing CpG sites with significant prognosis, as described above. Matrix A is decomposed into two non-negative matrices, W and H (i.e., A ≈ WH). Factorization of matrix A was repeated, its output was aggregated, and a consensus clustering of endometrial cancer samples was obtained. The optimal number of clusters was selected according to cophenetic, dispersion, and silhouette cophenetic.

### 2.4. Survival and Clinical Feature Analysis

Survival analysis was performed using the “survival” package in the R software (version 4.1.1). Survival rates of clusters of endometrial cancer defined by DNA methylation profiles were demonstrated using Kaplan–Meier plots. The log-rank test was used to determine the significance between clusters. All statistical tests of *p* < 0.05 are considered significant unless otherwise noted.

### 2.5. Specific DNA Methylation Markers for Endometrial Carcinoma Subgroups

Quantitative Differential Methylation Regions (QDMR) is an effective tool for quantifying methylation differences and identifying DMRS in multiple samples [22]. In this study, the quantitative method of QDMR was used to quantify methylation differences and to identify DMRs from genome-wide methylation profiles by adapting to Shannon entropy, which was used to find specific hypermethylation or hypomethylation CpG sites in specific endometrial cancer clusters. Finally, the classified sample with the largest absolute value of a specific cluster in each sample was determined to be the specific cluster corresponding to the specific CpG sites.

### 2.6. Gene Enrichment Analysis

In order to determine the biological function of gene enrichment by cluster, we downloaded pathways such as GO, KEGG, Hallmark, and Immunologic signatures and their gene sets in the MSigDB database. Next, gene collection enrichment analysis (GSEA) was performed on the specific gene sets of each cluster using the “fgsea” package in the R software. At the threshold of significance, which was a *p*-value less than 0.05, the biological functions of each cluster were significantly enriched.

### 2.7. Constructing a Predictive Model and Model Evaluation Based on Bayesian Network Classification

We constructed a Bayesian network classification model based on the train set. The samples in the test set were assigned to the corresponding subgroups using this classification model. The accuracy of the Bayesian network classification was evaluated with the “pROC” package in the R software. 

### 2.8. Immune Characteristics

CIBERSORT algorithms typically consider 22 immune cell types, such as B cells, T cells, NK cells, and macrophages, and are downloaded from the CIBERSORT website (http://CIBERSORT.stanford.edu/, accessed on 9 December 2021). We used the CIBERSORT algorithm to calculate the differences between 22 different immune cell types across clusters. The ssGSEA algorithm is used to study 28 kinds of immune cells, such as tumor cells with inhibition and killing properties. We obtained 28 immune enrichments based on the ssGSEA algorithm. The immune score, matrix score, and tumor purity in each cluster were calculated using the “estimate” package in the R software. The expression of immune checkpoints PD1, PDL1, PDL2, and CTLA4 in each cluster were considered. All used the Kruskal–Wilcoxon rank and test method, comparing the immune function of three clusters.

### 2.9. Predict Drug Sensitivity

We selected three common endometrial cancer chemotherapy drugs, including Cisplatin, Paclitaxel, and Docetaxel. Using the “pRRophetic” package in the R software, the half maximal inhibitory concentration (IC50) of each cluster of chemotherapy drug was calculated, which is a value that indicates the effectiveness of the substance in inhibiting a particular organism or biochemical process. Next, we assessed the relevance of three drugs to each cluster using the *t*-test method. 

## 3. Results

### 3.1. Prognostic Typing Based on DNA Methylation Characteristics

Clustering was performed using endometrial cancer DNA methylation profiles from the UCSC Xena database. First, we performed deletion of values for the data, removal of sex chromosomes and single nucleotide polymorphisms, and extraction of CpGs from promoter regions. The samples are then divided into train sets and test sets, with methylation data having the same expression profile. After the difference analysis of the data, 31,646 differential CpG sites were obtained. Based on 31,646 differential CpG methylation levels and survival information of cases, a univariate COX proportional risk regression model was established for each differential CpG site in the train set (containing 279 tumor samples). Through this analysis, 4898 CpG sites had significance (*p* < 0.05); that is, they had an impact on the survival of patients. Age (*p* = 0.0139) was an important factor influencing prognosis. Second, significant CpGs were placed in multivariate COX proportional risk regression models with age as covariates, in order to look for independent prognostic factors. Finally, 4870 CpG sites were significant and served as final classification features (Appendix A).

### 3.2. NMF Clustering of Endometrial Carcinoma Identified Distinct DNA Methylation Prognosis Subgroups

We performed unsupervised clustering based on β values of 4870 independent prognostically relevant CpG sites in order to obtain different molecular clusters of endometrial cancer DNA methylation prognosis. In order to determine the appropriate number of cluster classes, cophenetic, dispersion, and silhouette cophenetics were used to determine the optimal k-value, and k = 3 was selected as the optimal number of clusters after comprehensive consideration (Figure 1A). As shown in Figure 1B, when k = 3, we can see that the three clusters have clear boundaries, indicating that endometrial cancer samples have similar consistency. Heat maps were generated corresponding to dendrograms using “pheatmap” R packets, annotated with DNA methylation classification, age, clinicopathological staging, and grading (Figure 1C).

We then compared the prognostic differences between these three clusters. Kaplan–Meier survival analysis showed that the results between these groupings were statistically significant (Figure 2A). According to the endometrial cancer tumor grade (Figure 2B), in cluster 2, the proportion of grade I is the largest and the proportion of high grade is the lowest. In cluster 3, high grade has the highest proportion and grade I had the lowest. According to the clinicopathological stage of endometrial cancer (Figure 2C), there were no stage IV cases in subtype 2, and stage I accounted for the largest proportion. This clinical information suggests that of the subtypes we distinguished, cluster 2 had the best prognosis, cluster 1 was second, and cluster 3 had the worst prognosis. As can be seen from Figure 2D, the mean age of cluster 2 was the lowest, cluster 1 was second, and cluster 3 was the highest compared to other subtypes.

Next, we cross-compared TCGA molecular subtypes (TCGA cluster) in order to estimate whether our DNA methylation clusters were similar or different, corresponding to the same TCGA molecular subtypes. We analyzed DNA methylation clusters, enriching isotypes of the same TCGA molecule. CN high (Cluster 2 and Cluster 3) had significantly different methylation levels (Figure 1C). Among the subtypes we identified, CN low and MSI accounted for the largest proportion in cluster 2 (Figure 3E), and CN high accounted for the largest proportion in cluster 3 (Figure 3F), which is consistent with previous studies. CN low mutations lead to a good prognosis, and CN high gene mutations lead to a poor prognosis. Subsequently, we performed a log rank test on each pair of subgroups, and the results showed that the prognosis between cluster 1 and cluster 2 and between cluster 2 and cluster 3 was statistically significant (Appendix A). In addition, we used the transcriptome data corresponding to the subtypes in order to analyze the differences using the “limma” R packages. The “ggVennDiagram” R packages generated a Wayne map, found 1519 differential genes (Figure 2G), and used the “pheatmap” R packages to generate the corresponding heat map (Figure 2H). The expression levels were different between the clusters. It was indicated that different methylation sites have different differential expressions.

### 3.3. Specific Marker for the Prognostic Subgroups of DNA Methylation in Endometrial Cancer

In order to identify specific hypermethylation and hypomethylation CpG sites for DNA methylation clusters of endometrial cancer, we used QDMR software as a quantitative tool. A total of 4870 CpG sites were used for 3 clusters to look for specific CpG sites in each cluster. For each cluster, the average DNA methylation level per sample in 4870 CpGs was calculated, and 4870 three-dimensional matrices were inputted into the QDMR software. In order to find the specific CpGs site for each cluster, we lowered the threshold for the SD parameter to 0.03. Finally, 246 high/hypomethylation sites corresponding to 161 genes (Appendix A) were identified. These methylation sites were all DNA methylation markers of different clusters. The results showed that the number of specific CpGs in each cluster ranged from 32 to 120 (Figure 3A,B, Table 1), and that cluster 3 had the most specific methylation sites and contained the most hypomethylation sites. Subsequently, we performed Pearson correlation analysis on DNA methylation data and transcriptome data on these specific genes, and found that there were 55 related genes. Among them, the gene with the strongest negative correlation was *DLX5* (Figure 3C) and the gene with the strongest positive correlation is *SORCS2* (Figure 3D). Huang et al. found that DLX5 and TP63 had a synergistic effect to improve the activity and migration of squamous cell cells, thus promoting cancer progression [23]. Zhang et al. found that *DLX5* promotes the progression of osteosarcoma by activating the NOTCH signaling pathway [24]. Kaplan et al. reported that genetic variants located near *SORCS2* showed a significant genome-wide association with circulating concentrations of IGFBP-3 and IGF-I [25].

In addition, we performed KEGG, Hallmark, GO, and Immunologic signatures functional enrichment analyses for genes corresponding to specific CpG s in each DNA methylation cluster. The important pathways for each cluster are shown. Due to the small number of cluster 1 genes, fewer pathway types are enriched (Appendix A). The results showed that the specific genes in cluster 1 were mainly involved in the systemic lupus erythematosus pathway, the genes in cluster 2 were mainly involved in processes such as inflammatory response and cell matrix adhesion (Figure 4A), and the genes in cluster 3 were mainly involved in processes such as the p53 signaling pathway, bile metabolism, and protein modification (Figure 4B). This means that different DNA methylation clusters are involved in different functional and biological pathways.

### 3.4. Construction and Evaluation of Prognostic Models

In order to determine the discriminative ability of specific CpG sites to each cluster, we constructed a Bayesian network classifier model with a train set, and classified Bayesian networks with 246 specific methylation features. The model performance was evaluated by 10× cross-validation, and the classification accuracy reached 85.11% (Table 2). The area under the receiver operating characteristic curve (ROC) reached 0.863 (Figure 5A).

We then used this predictive model to predict the cases in the test set. The samples in the test set were assigned the categories corresponding to the train set. Survival analysis was performed on three clusters in the test set, and the results showed that they had statistically different prognoses (Figure 5B). This suggests that the specific CpGs in this study can be used as biomarkers for the prognosis of endometrial cancer. We performed the same clinical information analysis of the test set as the train set (Figure 5C,D). Subsequently, we also compared our classification assay with TCGA molecular typing. We used the same method as the train set and obtained consistent results (Figure 5E). In particular, CN low and MSI accounted for the largest proportion of subtype 2, as predicted in the test set. These results further illustrate the prediction accuracy of our model and the stability of its features. In addition, to verify that the clusters in the test set were similar to those in the train set, we compared clusters of the same marker in the train set and the test set, with no significant differences between the groups (Appendix A). These results suggest that cases predicted to belong to the same DNA methylated cluster have the same prognosis.

### 3.5. Immune Function and Tumor Microenvironment of Different Subgroups

The tumor microenvironment (TME) is essential for immune function, and has multiple clinical implications for the treatment of tumors. Next, we explored differences in immune cell infiltration in three clusters. Exploring the immune cell abundance of clusters using the CYBERSORT algorithm found that the proportion of T cells and macrophages in each cluster was relatively high (Figure 6A), which could mean that T cells and macrophages are potential targets for endometrial cancer treatment. The infiltration levels of 28 immune cells per cluster were assessed by using the ssGSEA algorithm (Figure 6B). We found that levels of activated B cells, CD4 T cells, CD8 T cells, immature B cells, and regulatory T cells were higher in cluster 1, levels of CD56dim natural killer cells and central memory CD4 T cells were lower in cluster 2, and levels of activated CD8 T cells, MDSCs, mast cells, neutrophils, and natural killer cells were lower in cluster 3. In addition, we calculated the immune score, stromal score, and tumor purity for each cluster using the ETIMATE algorithm (Figure 6C). The results showed that cluster 1 had the highest immune score and matrix score, followed by cluster 2, and cluster 3 had the lowest. Tumor purity showed the opposite trend, and was highest in cluster 3. These results suggest that patients with cluster 1 may benefit from immune checkpoint inhibitor therapy.

In recent years, immune checkpoint blockade has become the benchmark for many tumor treatments. Therefore, we explored the expression levels of some immune checkpoints, including PDCD1 (PD1), CD274 (PDL1), PDCD1LG2 (PDL2), and CD152 (CTLA4). As shown in Figure 6D, we found that cluster 1 had the highest levels of expression in PD1, PDL1, PDL2, and CTLA4, indicating that high expression of immune checkpoint inhibitors in cluster 1 patients may form an immunosuppressive microenvironment, further leading to tumor escape.

### 3.6. Prognostic Clusters Predict Drug Sensitivity

For advanced patients, chemotherapy drugs are the usual treatment for endometrial cancer. Therefore, we used the “pRRophetic” R package to predict drug sensitivity to three common chemotherapy drugs based on prognostic clusters. As shown in Figure 7A–C, these three chemotherapeutic drugs, namely Cisplatin (*p* = 0.00092), Paclitaxel (*p* = 0.0055) and Docetaxel (*p* = 5.1 × 10^−5^), have lower IC50 levels in cluster 3, indicating that patients in cluster 3 may respond well to chemotherapeutic drugs. For each cluster, Docetaxel had the lowest IC50 level (Figure 7D–F), indicating the best chemotherapy sensitivity. In addition, we evaluated the correlation of these three drugs with each cluster, and the results showed a positive correlation (Appendix A).

## 4. Discussion

Endometrial cancer is the most common gynecological cancer in developed countries and the sixth most common cancer in the world, with increasing morbidity and mortality worldwide [26]. Therefore, in order to improve the survival time of patients, there is an urgent need to classify endometrial cancer to identify new diagnostic biomarkers and to find new therapeutic targets. There is a potential link between epigenetics and cancer [27]. In some cancers, methylation genes are used as biomarkers for early detection, predicting cancer recurrence and response after various treatments. DNA methylation in the promoter region of tumor suppressor genes is an important mechanism in tumorigenesis. Among them, abnormal methylation has also been reported to be involved in the early changes in tumors, and plays a key role in the occurrence and development of cancer [28]. For example, HMLH1 promoter methylation is involved in early changes in the normal endometrium to carcinogenesis process, and is characteristic of a subset of precursor lesions [29]. Abnormal methylation of CpG in the promoter regions of GSTP1 and RASSF1A tumor suppressor genes is also an important event in the carcinogenesis process of endometrial cancer, which may affect the invasiveness of the tumor [30].

Whole genome bisulfite sequencing is the best method to study DNA methylation, but it has high analysis difficulty and high cost. The UCSC database is a publicly available resource covering various types of cancer omics. The Infinium HumanMethylation450 bead chip array dataset for endometrial cancer was used for our classification analysis. DNA methylation arrays are a good option for studying the methylation of whole-gene DNA in a large number of tumors. In this study, we obtained a new endometrial cancer classification based on DNA methylation data for endometrial cancer. We first selected prognostically related CpG sites in the promoter region for cluster analysis, and obtained three different prognostic subgroups by non-negative matrix decomposition clustering. This showed clinical differences and molecular differences, confirming the heterogeneity of endometrial cancer. In these three clusters, we found similarities with the TCGA molecular subtype. Subsequently, we identified 246 hypermethylation/hypomethylation sites, corresponding to 161 genes, and these genes were correlated in methylation data and transcriptome data. These sites can be used as targets for precision medicine and biomarkers for diagnosing endometrial cancer.

TME not only plays an extremely significant role in the occurrence, development, and metastasis of tumors, but also has a profound influence on the therapeutic effect [31]. Therefore, we investigated the relationship between subtypes and TME and immune cell infiltration. We found that patients with subtype 1 had higher levels of CD4 T cells and CD8 T cell infiltration than patients with other subtypes. Most clinically used drugs stop the mechanisms that dampen immune response; these drugs are called immune checkpoint inhibitors (ICIs). Among gynecological diseases, ICIs are one of the most effective methods to treat endometrial cancer [32]. In this study, we compared the expression levels of immune checkpoints between the three clusters. We found that these immune checkpoint inhibitors were expressed at higher levels in cluster 1. Dunn, G.P. et al. have shown that patients with high levels of gene expression at immune checkpoints have the potential to form an immunosuppressive microenvironment and promote tumor immune escape [33]. These results suggest that patients with cluster 1 may benefit from immune checkpoint inhibitor therapy. In addition, we predicted that cluster 3 might be more sensitive to commonly used chemotherapy drugs.

Although methylation may play an important role as a biomarker in endometrial cancer, the genes whose methylation sequences are affected in the promoter region remain unknown. Firstly, there is a lack of external datasets to further validate the prognostic prediction model. Second, in practice, the establishment of prediction models is much more complex and requires a variety of tools. In conclusion, based on the TCGA database and a series of bioinformatics methods, we have identified prognostic specific methylation sites and constructed a prognostic prediction model for endometrial cancer patients. The prognostic model we have constructed can provide clinicians with effective help and guidance on prognosis, clinical diagnosis, immune function, and medication strategies for patients with different subtypes of endometrial cancer.

In summary, using endometrial cancer data from the TCGA database, this study identified three different prognostic subtypes that differ at the molecular level, as well as clinical information that explains the heterogeneity of endometrial cancer. Specific CpG sites and genes in specific subsets can serve as biomarkers for personalized treatment, and clinicians can develop new treatment options based on these markers.

## Figures and Tables

**Figure 1 genes-13-02106-f001:**
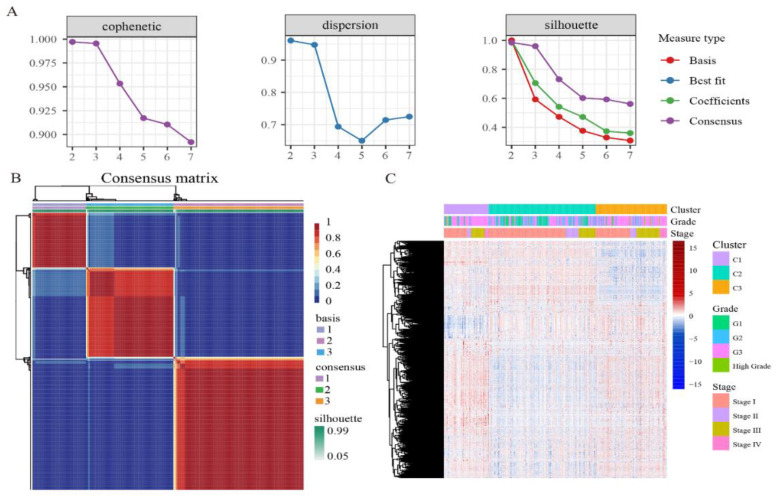
The NMF clustering for DNA methylation classification with the corresponding heat map. (**A**) Genetic correlation coefficients, dispersion, and silhouettes of cluster numbers 2 to 7. (**B**) NMF heat map at k = 3. (**C**) Corresponds to the heat map of the dendritic diagram in (**B**), annotated with clinicopathological staging and grading, and DNA methylation classification.

**Figure 2 genes-13-02106-f002:**
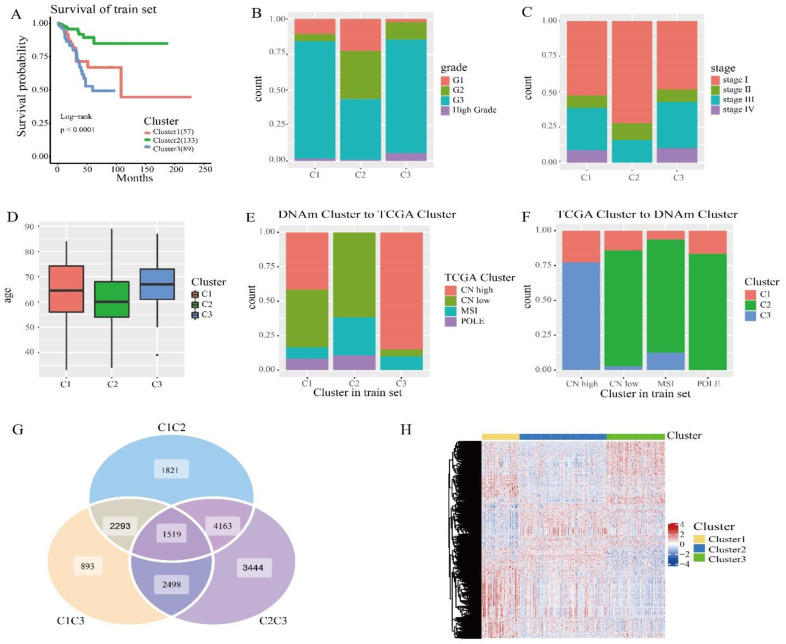
Survival curve of DNA methylation cluster, and comparison of DNA methylation cluster and its clinicopathological grading, stage, and age. (**A**) Survival curve of DNA methylated clusters in the train sets. The horizontal axis represents the survival time (months) and the vertical axis represents the survival probability. The numbers in parentheses in the legend represent the number of samples in each cluster. The logarithmic rank test was used to assess the statistical significance of the difference. (**B**) Clinicopathological staging, enriched in each DNA methylate cluster. (**C**) Clinical grading, enriched in each DNA methylate cluster. (**D**) Age distribution of 3 DNA methylated clusters in the train group. The transverse axis represents the DNA methylated cluster. (**E**) TCGA cluster with enrichment in each DNA methylation cluster. (**F**) The reverse orientation of (**E**). (**G**) Differentially expressed genes between clusters in the train sets. (**H**) Heatmap of differentially expressed genes in the train sets.

**Figure 3 genes-13-02106-f003:**
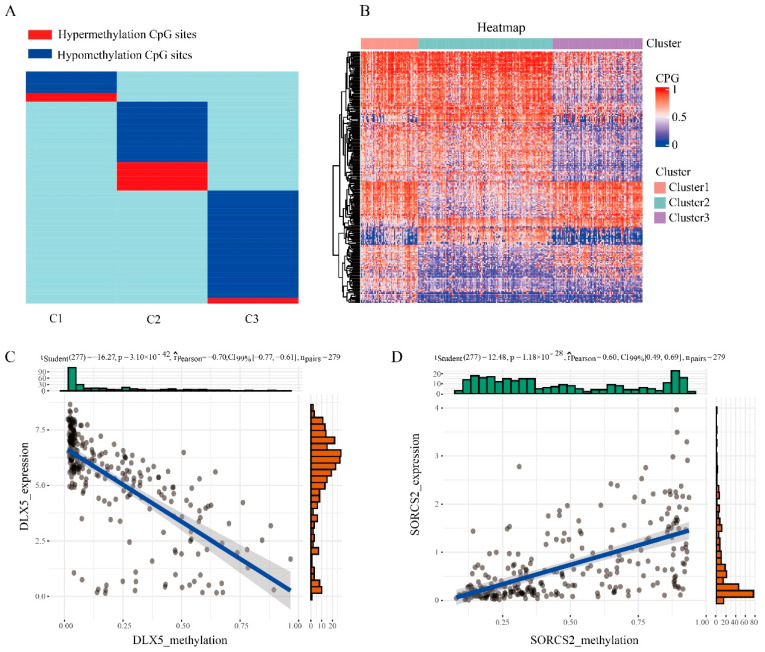
Specific hyper/hypomethylation CpG sites for each DNA methylation cluster. (**A**) Specific CpG sites for each DNA methylation prognostic subtype. The red bars and blue bars represent hypermethylation CpG sites and hypomethylation CpG sites, respectively. (**B**) Heat map of specific sites in 3 DNA methylation clusters. (**C**) The expression data and methylation data of the *DLX5* gene were negatively correlated. (**D**) The expression data and methylation data of the *SORCS2* gene were positively correlated.

**Figure 4 genes-13-02106-f004:**
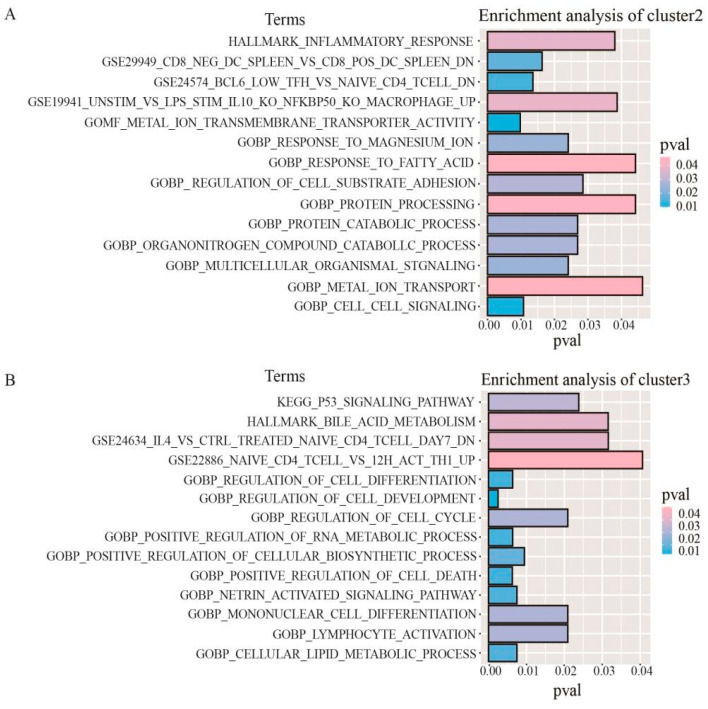
Gene enrichment analysis corresponding to specific CpG sites in DNA methylation clusters. (**A**) The enrichment analysis of genes corresponding to the specific CpG sites in cluster 2. (**B**) The enrichment analysis of genes corresponding to the specific CpG sites in cluster 3.

**Figure 5 genes-13-02106-f005:**
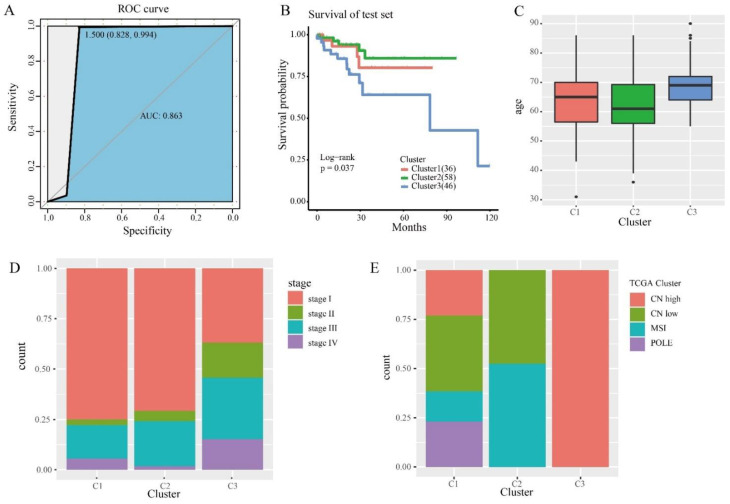
The prognostic model and prediction results. (**A**) The ROC curve shows the sensitivity and specificity of the prognostic model. The area under the curve (AUC) reached 0.863. (**B**) Survival curves of 3 clusters were predicted from the test set using the prognosis model. The log-rank test was used to assess the statistical significance of the difference (*p* = 0.037). (**C**) Age distribution of the three DNA methylation clusters in the test set. The horizontal axis represents DNA methylation cluster. (**D**) Clinical staging, enriched in each DNA methylation cluster. (**E**) TCGA cluster with enrichment in each DNA methylation cluster.

**Figure 6 genes-13-02106-f006:**
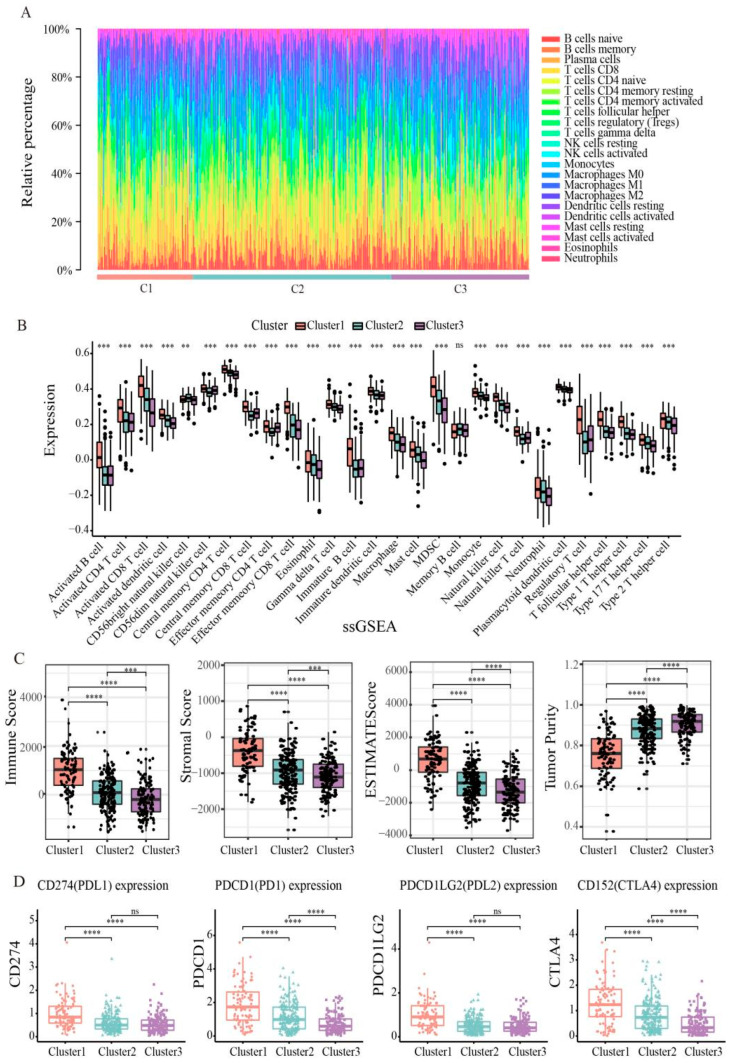
Immune function of endometrial cancer subtypes. (**A**) Box plot showing the distribution of tumor-infiltrating immune cells among clusters. (**B**) Box diagram of 28 immune cells of three clusters in endometrial cancer patients. (**C**) Three clusters of endometrial cancer patients’ immune scores, stromal scores, ESTIMATE scores and tumor purity. (**D**) Expression of immune checkpoints PD-1, PD-L1, PD-L2, and CTLA4 in three clusters. Statistical significance at the level of ns ≥ 0.05, ** < 0.01, *** < 0.001 and **** < 0.0001.

**Figure 7 genes-13-02106-f007:**
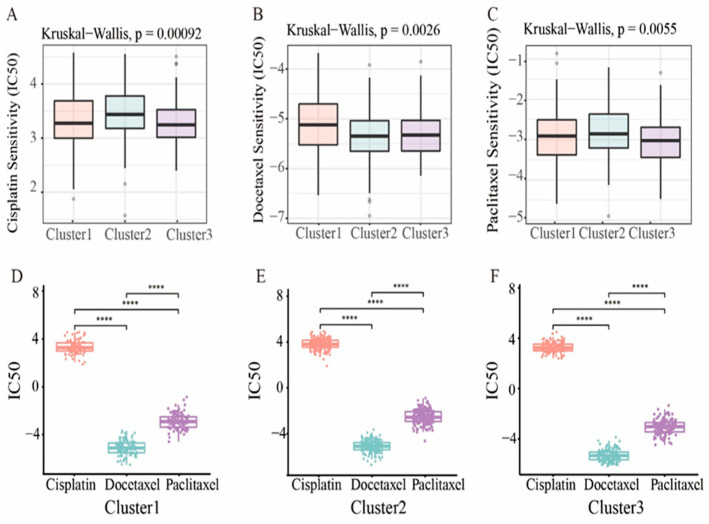
Drug sensitivity. (**A**–**C**) The boxplot showed the average difference in IC50 estimates for three representative drugs (Cisplatin, Docetaxel, and Paclitaxel). (**D**–**F**) Three of the subtypes represent the IC50 of the drug. Statistical significance at the level of **** < 0.0001.

**Table 1 genes-13-02106-t001:** The numbers of specific CpGs for clustering.

Cluster	Number of Specific CpGs
Cluster 1	32
Cluster 2	94
Cluster 3	120

**Table 2 genes-13-02106-t002:** The confusion matrix of Bayesian network classification. Each row of the matrix represents the instances in the prediction cluster, and each column represents the instances in the actual cluster. Classification accuracy reached 85.11%.

	Cluster 1	Cluster 2	Cluster 3
Cluster 1	27	10	0
Cluster 2	3	49	6
Cluster 3	2	0	44

## Data Availability

Support for the conclusion of the original data can be found at https://xenabrowser.net/datapages/ (accessed on 21 January 2022).

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
