# Peer review of "Excavation of Molecular Subtypes of Endometrial Cancer Based on DNA Methylation"

_genes, 2022, doi:10.3390/genes13112106_

Round 1
Reviewer 1 Report
Liu et al. titled by “Molecular subtypes based on DNA methylation predict prognosis in endometrial cancer patients” identified 3 distinct clusters of endometrial cancer patients based on DNA methylation pattern in TCGA database. Based on characteristics of each cluster, the authors suggested that patients in each cluster might get benefit from different therapeutic strategies. It is interesting to see three distinct clusters of endometrial cancer patients based on DNA methylation, which enabled to stratify patients with differential prognosis. However, it is not clear whether the prognostic model has been validated using an independent dataset. Since this entails re-analysis of the TCGA dataset, it is critical to provide what the authors add on top of the previous findings, which is lacking in the manuscript. Furthermore, potential clinical benefits for immune checkpoint inhibitors and chemotherapy based on DNA methylation pattern are highly speculative, which requires further confirmation or the authors need to provide more convincing evidence. Overall, it is not clear what novel findings have been added from the previous TCGA paper.
Major points
In the previous TCGA paper, four distinct clusters (MC1-4) of endometrial cancer have been reported. It is important for the authors to cross-compare if there is any similarity or differences, which might shed a new light on the molecular subtype. For instance, MC1 is a heavily methylated subtype reminiscent of the CpG island methylator phenotype (CIMP) described in colon cancers and glioblastomas. Does MC1 overlap with the cluster 1 in this analysis? Another example, MC3 has frequent mutations in TP53, which might get benefit from chemotherapy.
TCGA data also include transcriptomic data associated with patients. The authors could have provided comprehensive analysis whether differentially methylated regions/sites can result in differential gene expression. Also it is interesting to see 246 methylation sites are heavily enriched in CpG islands or gene bodies or intergenic regions.
The authors used the train set and the test set for patients’ prognosis. It is not clear which dataset was used for the train set and the test set. TCGA data was used for the train set, and other independent data set was used for the test?
The authors suggested that patients in cluster 1 can get benefit from immune checkpoint inhibitor therapy, and cluster 3 from common chemotherapies. To this end, the authors compared it with another prediction algorithm. Still it is highly speculative, which requires further validation with retrospective data or any other means. This should be tone down significantly throughout the manuscript or should be supported by further evidence.
Minor points
‘Hypermethylation’ and hypomethylation terms need to be consistently used throughout the manuscript. (e.g., line 20, high and hypomethylation)
Reviewer 2 Report
The Yujie et al. reported the molecular subtypes of DNA methylation for predicting prognosis in endometrial cancer. DNA methylation will affect gene expression and heterogeneity. The authors analyzed 482 endometrial cancer methylation data of TCGA, and distinguished 3 important clusters from 4870 clusters. The high and hypomenthylation sites were identified and it constructed a prognostic model for Bayesian network classification. The clusters from DNA methylation data was used to determine the immune function of each cluster, and predict the sensitivity of chemotherapy. This study may provide important DNA methylation information to address the heterogeneity and lay the ground of effective and individualized treatments.
There are some specific concerns as follow,
1, The color of basis and consensus is difficult to distinguish the clusters in the Figure 1B. Perhaps, it is better to make it bigger or showed the color in a better way.
2, The figure 1A, the last figure lost 7, because there are 23456.
3, The figure 4 and figure6 are not clear. The figure is under shadow somehow, and it is better to improve figure resolution.
4, The author mentioned DNA methylation sites of gene promoter. Does the number of methylation sites in one promoter affect the gene expression? For example, some promoters include several methylation sites but some promoters have one methylation site.
5, How accurate the prognostic prediction model is for endometrial cancer prognosis, clinical diagnosis and immune function? Because there are many factors affect the endometrial cancer and methylation is one of cancer factors.
6, There are cluster1, cluster2, cluster3 for endometrial cancer and each cluster contain many methylation sites. If the patients methylations are between the cluster1, cluster2, cluster3, How to distinguish the patient cancer by this analysis?

Round 2
Reviewer 1 Report
The authors addressed the major concerns and revised the manuscript accordingly.